# Implementation of Medical Hospitalist Care at a Korean Tertiary Hospital: A Retrospective Cross-Sectional Study

**DOI:** 10.3390/jcm13216460

**Published:** 2024-10-28

**Authors:** Han Sung Lee, Seung Kyo Park, Sung Woo Moon

**Affiliations:** Division of Integrated Medicine, Department of Internal Medicine, Yonsei University College of Medicine, Seodaemun-gu, Seoul 03722, Republic of Korea; hansunglee@yuhs.ac (H.S.L.); seungkyo_park@yuhs.ac (S.K.P.)

**Keywords:** hospitalist, hospital medicine, treatment outcome

## Abstract

**Background/Objectives**: In March 2018, a tertiary teaching hospital launched a medical hospitalist team. This study presents the clinical characteristics and outcomes of medical hospitalist care and reveals the relationship between them. **Methods**: This study included 4003 patients first admitted to the hospitalist team via emergency room and then discharged from the hospitalist team between March 2018 and November 2022. The patients were admitted either to the teaching admitter hospitalist team or the hospitalist-led acute medical unit (AMU). Afterward, the patients were either discharged, if possible, within a few days or transferred to ward hospitalists if assigned wards for hospitalist care were available. **Results**: The patients’ mean Charlson Comorbidity Index score was 3.5 and the mean National Early Warning Score was 3.4. Of the admissions, 44.2% of the patients were admitted to the AMU, and 26.8% received an early consultation with a subspecialist. Each hospitalist managed 12.8 patients per month on average. The patients’ mean LOS was 14.52 days, 10.5% of patients died during hospitalization, and 13.0% of patients had unscheduled readmission within 1 month. The patients’ mean total cost per hospital stay was 572,836 won per day. Admission to the AMU was associated with a lower total cost per hospital stay, but the relationships with mortality, readmission, and LOS were not significant. **Conclusions**: The study reports on the outcomes of implementing a medical hospitalist care system that combines short-term admission wards with integrated care models to manage complex cases. These findings provide insights into optimizing hospitalist systems for improved patient outcomes.

## 1. Introduction

The term “hospitalist” originated in the United States around 30 years ago, and there are now over 50,000 hospitalists in the United States [1]. Hospitalists are highly trained in managing complex medical conditions and provide around-the-clock care to patients [2]. Hospitalist care results in improved patient outcomes and increased efficiency [3,4,5]. While there has been keen interest in hospitalist care in Asia, its widespread adoption is lacking, and studies on hospitalist care in Asia remain insufficient [6].

In Korea, most inpatient care is handled by resident trainees, who subsequently face heavy workloads [7]. The need for hospitalists in Korea has emerged due to regulatory changes that have limited working hours for resident trainees, reductions in the training period for internal medicine, and safety issues with inpatient care [8,9]. The hospitalist system was first implemented in Korea as a pilot project in September 2016 to improve the safety and quality of medical care for inpatients. Based on the clinical results from the pilot project, the government promoted the hospitalist system to a regular service in January 2021 and established a new fee schedule [10]. Accordingly, monetary compensation has improved for hospitalists [11], with 304 hospitalists currently working in Korea [10].

There are three different hospitalist ward models in the Korean healthcare system: the general ward model, the short-term admission ward model, and the integrated care model [7]. In the general ward model, patients admitted to hospitalists are limited to designated subspecialties; moreover, depending on the hospital, the hospitalists manage patients under the direction of subspecialists [7]. In the short-term admission ward model, patients have a typical turn-around period of 72 h [7]. In the integrated care model, patients from all specialties are admitted to the same ward; therefore, hospitalists care for patients independently [7].

In March 2018, a tertiary teaching hospital in Korea (Severance Hospital) launched a medical hospitalist team to implement medical hospitalist care. The medical hospitalist team was composed of internal medicine board-certified physicians and was committed to treating patients from the emergency room (ER). Through trial and error, the team established a hospitalist system that combined the short-term admission ward and integrated care models. Under this system, patients were first admitted to the short-term admission hospitalists (i.e., the admitter teaching hospitalist and internal medicine resident team or the hospitalist-led acute medical unit (AMU)) under the short-term admission ward model and were then transferred to ward hospitalists, who managed patients under the integrated care model, if they were not discharged within a few days. From the data accumulated from this experience, this study aims to report: (1) the clinical characteristics and outcomes of medical hospitalist care, and (2) the relationship between the clinical characteristics and outcomes based on the medical hospitalist team’s initial experience at a tertiary hospital in Korea.

## 2. Materials and Methods

### 2.1. Establishment and Processes of the Medical Hospitalist Team

Figure 1 shows the change in the number of hospitalists, hospitalist-assigned wards, mean number of admissions per month, and the number of admissions per month divided by the number of hospitalists. After the establishment of the medical hospitalist team in March 2018, two medical hospitalists, one internal medicine resident, and one ward were assigned for hospitalist care. The medical hospitalist team took care of patients admitted via the ER. By December 2020, one additional ward was assigned, and the AMU was launched, compromising two hospitalists who worked for 10 h a day, 7 days a week, on alternate weeks. By June 2021, one additional ward was assigned to the medical hospitalist team. From March 2018 to November 2022, the medical hospitalist team recruited 17 internal medicine boarders with subspecialties (five from nephrology; three each from cardiology, pulmonology, and infectious diseases; and one each from gastroenterology, oncology, and endocrinology), among whom seven resigned.

Figure 2 shows the flow of patients admitted to the medical hospitalist team. As the hospitalist team was unable to handle all ER patients, the criteria for admission to the medical hospitalist team from the ER were as follows: (1) if a patient had complex medical problems, or the major cause of the problem was not revealed during the initial evaluation at the ER; (2) if there were legal issues or a guardian was absent; (3) if hospitalization was a bridge to ICU admission; and (4) if there was a request from a specialty department. If a patient met the criteria for admission to the medical hospital team and the medical hospitalist team was capable of admitting patients, the patient was primarily admitted to the AMU. If the designated number of patients (3–4 admissions per day) was met, the patients were then admitted to the teaching admitter hospitalist and internal medicine resident team. They then went through the following process: discharged by the admitter teaching hospitalist or the AMU if they could be discharged within a few days, transferred to ward hospitalists if assigned wards for hospitalist care were available and early discharge was impossible, or transferred to specialist departments if specialist care was needed. If a patient went to the ICU during admission, they were transferred to the critical care team; if a patient left the ICU, they were transferred back to the hospitalist team. The medical hospitalist team provided patient coverage exclusively on workdays, with on-call doctors managing patients on non-workdays, while the AMU team ensured continuous coverage throughout the entire year, including all 365 days.

### 2.2. Study Design

This study was a retrospective cross-sectional descriptive study conducted at a tertiary teaching hospital in Korea. The study population included patients admitted to the medical hospitalist team from the ER between March 2018 and November 2022. Among the 5435 admissions by 4705 patients to the medical hospitalist, only the first admissions were included in the study, as readmission was one of the main outcomes of interest. This study excluded 801 admissions by 702 patients who were transferred to specialist care. In total, this study included 4003 patients’ first admissions to the medical hospitalist team in the analysis.

This study collected data on the patients’ age, sex, BMI, smoking/alcohol-consumption status, underlying comorbidity (measured via the Charlson Comorbidity Index [CCI]), acute-illness severity (measured via the National Early Warning Score [NEWS] and Korean Triage and Acuity Scale [KTAS]), and admittance to a teaching admitter hospitalist; whether an early consultation with a subspecialist was conducted; hospitalists’ workload at the time of admission; major cause of admission; whether ICU admission, mechanical ventilation, or renal replacement therapy were required during admission; and patients’ length of stay (LOS) at hospital, type of discharge, unscheduled readmission within one month, total cost of hospital stay, and mean total cost of hospital stay per day. The CCI used in this study was calculated based on Charlson et al. [12]. The NEWS measured in this study was calculated using triage records following the 2012 report by the Royal College of Physicians [13]. Among the 4003 patients, NEWS scores were obtainable for 2648 individuals, excluding those directly referred from outpatient clinics to the emergency department without triage records and those whose critical condition precluded the completion of triage assessment. The KTAS was calculated using triage records following the report developed through the Korean Ministry of Health and Welfare’s research project [14] and was also available for 2648 of the 4003 patients. Early consultation with a subspecialist was defined as requiring a clinical consultation with an internal medicine subspecialist within 24 h of admission. Hospitalists’ workload was indirectly measured by the mean number of admissions/month/number of hospitalists at admission per quarter.

### 2.3. Statistical Analysis

Descriptive statistics were used to describe the variables under study with proportions or means with standard deviations and medians with interquartile ranges. Multivariate logistic regression models were constructed to identify the risk factors for planned discharge, in-hospital death, and unscheduled readmission within one month. Multivariate regression analyses were performed to identify the factors associated with LOS and the mean total cost per hospital stay. The covariates entered into the multivariate logistic regression analyses and multivariate regression analyses included age, sex, BMI, CCI, NEWS, admission via a teaching hospitalist, hospitalists’ workload, early consultation from an internal medicine specialist, and complex problems as a major cause of admission. The variables included in all multivariable analyses were tested for multicollinearity, while the KTAS was not included in the multivariate analysis due to multicollinearity with the NEWS. An adjusted *p*-value (<0.05) was considered statistically significant. All statistical analyses were performed using SPSS Version 26.0 (SPSS Inc., Chicago, IL, USA).

## 3. Results

### 3.1. Patients’ Baseline Characteristics and Outcomes

Table 1 shows the patients’ baseline characteristics. Patients’ mean age was 70.7 ± 15.5 years, mean CCI was 3.5 ± 2.3, and mean NEWS was 3.41 ± 3.07. Of the 4003 admissions, 44.2% were admitted through the AMU, and 26.8% had early consultations with internal medicine subspecialists. Each hospitalist managed a mean of 12.8 ± 2.6 patients per month. The most frequent cause of admission was pulmonary disease (32.1%). The final diagnoses of patients admitted for each major cause of admission are in Appendix A. Figure 1 shows the hospitalists’ workload per quarter each year.

Table 2 shows the medical-hospitalist care outcomes. The mean LOS was 14.52 ± 15.23 days, and 10.14% of patients’ LOS exceeded 1 month. Overall, 79.4% of patients had planned discharge, while 10.5% of patients died during hospitalization. The mean total cost per hospital stay was 572,836 ± 294,585 won per day. After discharge, 13.04% of patients had unscheduled readmission within 1 month.

### 3.2. Results of the Logistic Regression Analyses

Table 3 shows the relationship between the covariates and planned discharge, in-hospital death, and unscheduled readmission within one month in the respective multivariate logistic regression analyses. Age, BMI, CCI (OR: 0.897, 95% CI: 0.855–0.941, *p* < 0.001), and NEWS (OR: 0.871, 95% CI: 0.840–0.902, *p* < 0.001) were independently related with planned discharge. Age, BMI, CCI (OR: 1.189, 95% CI: 1.116–1.266, *p* < 0.001), NEWS (OR: 1.167, 95% CI: 1.114–1.221, *p* < 0.001), and nephrology problems as a major cause of admission (OR: 0.503, 95% CI: 0.276–0.919, *p* = 0.025) were independently related with in-hospital death. BMI, CCI (OR: 1.189, 95% CI: 1.116–1.266, *p* < 0.001), nephrology problems (OR: 0.643, 95% CI: 0.440–0.938, *p* = 0.022), and other diseases as a major cause of admission (OR: 0.538, 95% CI: 0.292–0.988, *p* = 0.046) were independently related to unscheduled readmission within one month. There was no correlation between admission to the hospitalist-run AMU, hospitalists’ workload, early consultations with internal medicine subspecialists, planned discharge, in-hospital death, and unscheduled readmission.

### 3.3. Results of the Multiple Regression Analysis

Table 4 shows the relationship between the covariates, LOS, and the total cost per hospital stay in the respective multivariate multiple regression analyses. CCI (*β* = 0.060, *p* = 0.005); NEWS (*β* = 0.130, *p* < 0.001); and pulmonology, hemato-oncology, and infectious disease as a major cause of admission were the main determinants for LOS, accounting for 72% of the variance. CCI (*β* = 0.060, *p* = 0.005); NEWS (*β* = 0.130, *p* < 0.001); admission to the hospitalist-run AMU (*β* = −0.066, *p* = 0.002); and hemato-oncology, nephrology, and other diseases as a major cause of admission were the main determinants for total cost per hospital stay, accounting for 70% of the variance.

## 4. Discussion

The application of hospitalists in Asia is still evolving as the specialty continues to develop. Owing to the relatively new concept of hospitalists as a distinct medical specialty, there is still much debate and experimentation regarding their work schedules and responsibilities. The field will continue to adapt and evolve in response to the changing needs of patients and healthcare systems [15].

The hospitalist system examined herein combined the short-term admission ward and integrated care models. The hospitalists worked in a rotation of two models according to their schedules. The short-term admission ward model promotes the efficient use of hospital beds without compromising patient outcomes [16]. The advantage of the integrated care model is that it gives full autonomy to the hospitalists, so their satisfaction levels are high [11]. However, in the current study, the hospitalists frequently reported burnout during the short-term admission ward rotation and were reluctant to work there.

A study in Japan reported that patients admitted to hospitalist care had a median CCI of 1.4 [17]. The hospitalist system emerged in Korea to relieve the burden of resident trainees and reduce the safety issues regarding inpatient care, as the LOS, complexity, severity, and mortality rates of patients admitted to the hospitalists’ care were high and comparable to the semi-ICU. A previous study in Korea reported that patients admitted to the ICU had a mean CCI of 3.0 [18], while the in-hospital mortality rate of patients after ICU admission was 12.1%. Meanwhile, a study in Sweden found that only 13.1% of patients who were admitted to the ICU had a CCI that exceeded 3.0 [19]. Our result shows that the mean CCI was 3.5, and 62.1% of patients had a CCI that exceeded 3.0. This may be beneficial for patients, as previous studies have reported that the more complex a patient’s problems, the more significant the impact of the hospitalists [17,20,21].

This study further highlights the importance of increasing the number of hospitalist-assigned wards. In the first quarter of 2022, the number of hospitalists increased from 7 to 12. Despite the additional recruitment of hospitalists, the admissions to the hospitalist team did not dramatically increase. In Korea, the number of hospitalists required remains insufficient, and many hospitals are struggling to recruit hospitalists [11]. This study found that while many hospitalists were recruited, the mean admissions per month did not increase as the number of assigned wards did not rise.

This study also found that the hospitalists’ workload (mean admissions/month/number of hospitalists at admission per quarter) was not related to any of the outcomes measured in the current study. Hospitalists’ high workload has previously been linked to worse patient outcomes [22,23,24]. However, as the patient safety considerations often require capping the maximum number of patients hospitalists can manage, and safety measures are not included in the current study, cautious interpretation is warranted when analyzing the results in this context. In the future, when the number of assigned wards increases and the number of patients rises, a re-evaluation of the relationship is required.

The results show that teaching hospitalist care has been associated with shorter LOS, with no adverse effects on readmission or mortality rates [25]. Comparing the teaching admitter hospitalist team and the hospitalist-run AMU in this study, the total cost of hospital stay per day was higher for the teaching admitter hospitalist team when adjusted for compounding factors, while the in-hospital mortality, readmission, and LOS rates did not differ between the two groups. One reason may be that the teaching admitter hospitalist team only covered workdays while the admitter hospitalist team ensured continuous coverage throughout the entire year. Some studies have reported the negative effect of weekday coverage on higher in-ward mortality, higher ICU admission, and lower LOS [9,26]. Another reason may be the involvement of internal medicine residents in treatment during training courses; some studies have reported a link between medical residents’ errors and worsened patient outcomes [27,28].

This study found that only 26.8% of patients had an early consultation with an internal medicine subspecialist, and early consultation with a subspecialist was not associated with outcomes when adjusted for compounding factors. This restriction of early consultations with subspecialists may be because all medical hospitalists were board-certified internal medicine doctors with subspecialties who could collaborate and conduct shared decision-making within the hospitalist team. Collaboration is a key element of patient management in the hospitalist team.

This study has the following limitations: First, the hospitalists’ workload was indirectly measured through the number of admissions per month divided by the number of hospitalists. Methods for measuring the hospitalists’ workload included measuring patient complexity using CCI, total electronic health record times, work spent outside of work, surveys, and so on [29,30]. Second, as aforementioned, the NEWS was not available for patients who were deemed too critical for evaluation at triage. This may have resulted in the underestimation of the NEWS, and the patients’ severity may have been higher in reality. However, the relationships between the NEWS and various outcomes were confirmed, which is consistent with the extant literature [31,32]. Third, other quality indicators, such as patient or staff satisfaction, were not evaluated herein. Fourth, the study was retrospective in nature and included patients from a single center.

## 5. Conclusions

This study evaluated the outcomes of implementing a medical hospitalist team to manage patients with complex conditions, severe diseases, and high mortality rates within a hospitalist system that combines short-term admission wards and integrated care models. Further, by examining various factors related to patient outcomes, this study aimed to understand how these factors affect the quality of care.

## Figures and Tables

**Figure 1 jcm-13-06460-f001:**
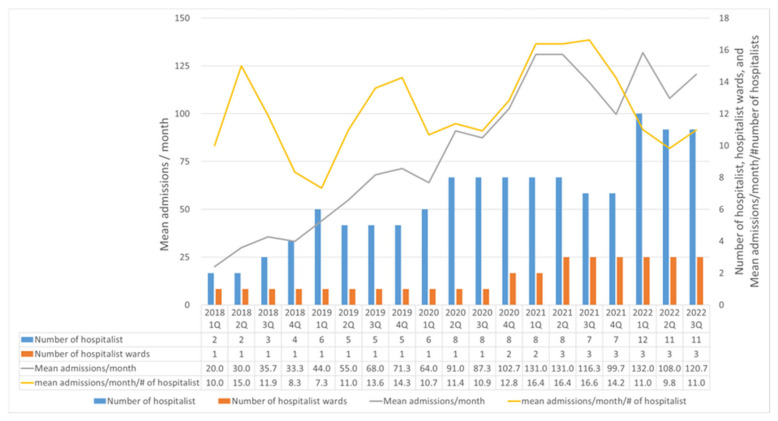
Change in Number of Hospitalists, Assigned Wards, and Admissions.

**Figure 2 jcm-13-06460-f002:**
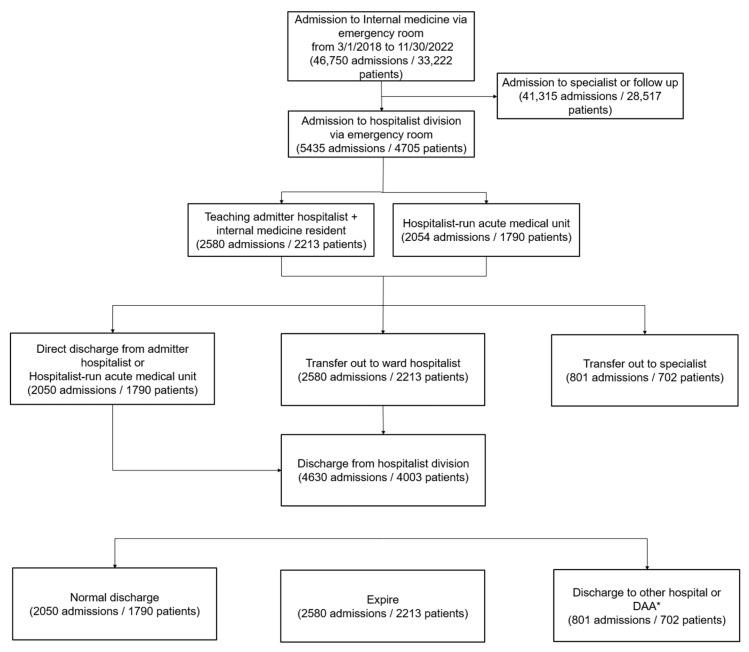
Summary of the flow of patients admitted to the medical hospitalist team. * DAA: discharge against advice.

**Table 1 jcm-13-06460-t001:** Participants’ Baseline Characteristics.

Variable	Mean ± SD or *n* (%)
Age, years	70.7 ± 15.5
Sex, female	1803 (45.0%)
BMI (kg/m^2^)	22.2 ± 4.1
Smoking, current, or ex-smoker	890 (22.2%)
Alcohol, yes	996 (24.9%)
Charlson Comorbidity Index	3.5 ± 2.3
0~2	1516 (37.9%)
3 or 4	1271 (31.8%)
≥5	1216 (30.4%)
National Early Warning Score (NEWS)	3.41 ± 3.07
Low (1–4)	1903 (71.9%)
Moderate (5 or 6)	336 (12.7%)
High (≥7)	409 (15.4%)
Admission via hospitalist-run AMU	1769 (44.2%)
Hospitalists’ workload ^1^	12.8 ± 2.6
Early consultation with an internal medicine subspecialist ^2^	1071 (26.8%)
Major cause of admission	
Gastroenterology	852 (21.3%)
Pulmonology	1286 (32.1%)
Cardiology	208 (5.2%)
Hemato-oncology	159 (4.0%)
Nephrology	786 (19.6%)
Infectious disease	368 (9.2%)
Other disease	344 (7.4%)
Korean Triage and Acuity Scale (KTAS) classification ^3^	
I	99 (3.59%)
II	520 (18.88%)
III	1368 (49.67%)
IV	690 (25.05%)
V	77 (2.80%)

^1^ Mean admissions/month/hospitalist at the time of admission. ^2^ Consultation with a subspecialist within 24 h of admission. ^3^ Acquired from 2648 of 4003 patients.

**Table 2 jcm-13-06460-t002:** Outcomes of Medical Hospitalist Admission.

Variable	Mean ± SD or *n* (%)
ICU admission during admission	147 (3.7%)
Mechanical ventilation during admission	171 (4.3%)
Renal replacement therapy during admission	400 (10.0%)
Length of hospital stay	14.52 ± 15.23
Length of hospital stay ≥ one month, *n* (%)	406 (10.14%)
Type of discharge	
Planned discharge	3180 (79.4%)
In-hospital death	419 (10.5%)
Hospital transfer	353 (8.8%)
Discharge against medical advice	51 (1.3%)
Unscheduled readmission within one month	522 (13.04%)
Total cost, won	8,509,775 ± 12,258,452
Total cost per hospital stay, won/day	572,836 ± 294,585

**Table 3 jcm-13-06460-t003:** Relationship Between Covariates and Outcomes in Logistic Regression Analyses.

Variable	Planned Discharge		In-Hospital Death		Unscheduled Readmission Within One Month	
	OR (95% CI)	*p*-Value	OR (95% CI)	*p*-Value	OR (95% CI)	*p*-Value
Age, years	0.978 (0.969–0.987)	<0.001	1.018 (1.005–1.031)	0.006	0.994 (0.986–1.002)	0.135
Sex, female	0.915 (0.722–1.158)	0.459	0.954 (0.689–1.321)	0.776	0.931 (0.725–1.196)	0.578
BMI (kg/m^2^)	1.096 (1.065–1.129)	<0.001	0.928 (0.891–0.966)	<0.001	0.945 (0.917–0.974)	<0.001
Charlson Comorbidity Index	0.897 (0.855–0.941)	<0.001	1.189 (1.116–1.266)	<0.001	1.079 (1.026–1.135)	0.003
National Early Warning Score (NEWS)	0.871 (0.84–0.902)	<0.001	1.167 (1.114–1.221)	<0.001	1.008 (0.966–1.051)	0.717
Admission via hospitalist-run AMU	1.009 (0.798–1.275)	0.943	0.833 (0.602–1.153)	0.271	0.953 (0.744–1.219)	0.700
Hospitalists’ workload ^1^	0.991 (0.948–1.035)	0.674	1.025 (0.965–1.089)	0.426	0.992 (0.948–1.039)	0.743
Early consultation with an internal medicine sub-specialist ^2^	1.109 (0.847–1.45)	0.452	1.043 (0.727–1.497)	0.819	0.891 (0.674–1.179)	0.419
Major cause of admission						
Gastroenterology	Reference	Reference	Reference	Reference	Reference	Reference
Pulmonology	0.787 (0.561–1.104)	0.165	1.346 (0.855–2.119)	0.199	0.826 (0.588–1.162)	0.273
Cardiology	1.656 (0.85–3.225)	0.138	0.9 (0.396–2.048)	0.803	0.893 (0.508–1.568)	0.693
Hemato-oncology	0.768 (0.407–1.45)	0.416	2.007 (0.944–4.269)	0.070	1.117 (0.622–2.006)	0.710
Nephrology	1.452 (0.975–2.163)	0.067	0.503 (0.276–0.919)	0.025	0.643 (0.440–0.938)	0.022
Infectious disease	0.831 (0.525–1.314)	0.429	0.614 (0.296–1.274)	0.191	0.652 (0.393–1.083)	0.098
Other disease	0.667 (0.401–1.109)	0.119	0.608 (0.256–1.446)	0.261	0.538 (0.292–0.988)	0.046

^1^ Mean admissions/month/hospitalist at the time of admission. ^2^ Consultation with a sub-specialist within 24 h of admission.

**Table 4 jcm-13-06460-t004:** Relationship Between Covariates and Outcomes in Multiple Regression Analyses.

Variable	Length of Hospital Stay (R^2^ = 0.72)		Total Cost per Hospital Stay (R^2^ = 0.70)	
	Standardized *β*	*p*-Value	Standardized *β*	*p*-Value
Age, years	0.028	0.186	0.003	0.891
Sex, female	0.018	0.397	−0.019	0.371
BMI (kg/m^2^)	−0.045	0.03	−0.022	0.277
Charlson Comorbidity Index	0.060	0.005	0.063	0.003
National Early Warning Score (NEWS)	0.130	<0.001	0.181	<0.001
Admission via hospitalist-run AMU	−0.037	0.078	−0.066	0.002
Hospitalists’ workload ^1^	−0.022	0.291	−0.008	0.7
Early consultation with an internal medicine subspecialist ^2^	0.031	0.135	0.091	0.891
Major cause of admission				
Gastroenterology	Reference	Reference	Reference	Reference
Pulmonology	0.098	0.001	0.003	0.907
Cardiology	0.024	0.288	0.045	0.046
Hemato-oncology	0.075	0.001	0.079	<0.001
Nephrology	−0.007	0.79	−0.059	0.023
Infectious disease	0.126	<0.001	0.014	0.561
Other disease	0.023	0.319	0.046	0.045

^1^ Mean admissions/month/hospitalist at the time of admission. ^2^ Consultation with a subspecialist within 24 h of admission.

## Data Availability

Data used and/or analyzed for this study are available from the corresponding author (S.W. Moon) on reasonable request.

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
