# Peer review of "Implementation of Medical Hospitalist Care at a Korean Tertiary Hospital: A Retrospective Cross-Sectional Study"

_jcm, 2024, doi:10.3390/jcm13216460_

Round 1
Reviewer 1 Report
Comments and Suggestions for Authors
- Line 125-127 (highlighted in PDF), this sentence feels incomplete. later on it is explained that the NEWS was not registered completely for patients deemed to critical at triag, but this does not reflect from this sentence.
- the 'protective' effect from several major causes of admissions might ask for more explanation. at line 176-180, nephrology problems is associated with OR 0.503 on in-hospital death. It might be interesting to include a table in supplements with reasons for admission within each major category as to understand why nephrology of 'other diseases' offer a protective effect against mortality or unscheduled readmissions.
- this could be further discussed in the discussion with some extra explanation.

Author Response
-
- Line 125-127 (highlighted in PDF), this sentence feels incomplete. later on it is explained that the NEWS was not registered completely for patients deemed to critical at triag, but this does not reflect from this sentence.
Reply: We sincerely apologize for the ambiguity in our original statement and any inconvenience this may have caused. We appreciate your valuable feedback and have revised the sentence for clarity. The updated version now reads:
'Among the 4,003 patients, NEWS scores were obtainable for 2,648 individuals, excluding those directly referred from outpatient clinics to the emergency department without triage records and those whose critical condition precluded the completion of triage assessment.'
We believe this revision provides a more precise and comprehensive description of our patient selection process. We thank you for your kind comment and the opportunity to improve the clarity of our manuscript. (Page 4, line 127-130)
- the 'protective' effect from several major causes of admissions might ask for more explanation. at line 176-180, nephrology problems is associated with OR 0.503 on in-hospital death. It might be interesting to include a table in supplements with reasons for admission within each major category as to understand why nephrology of 'other diseases' offer a protective effect against mortality or unscheduled readmissions.
Reply: Thank you for your valuable comment. As you pointed out, there were differences in final diagnosis for each major cause of admissions. To better illustrate this, we have added the final diagnoses of patients with ICD-10 codes for each major cause of admission in the supplementary table. We appreciate your insightful feedback. (Page 5, line 159-161, Supplementary materials)
Reviewer 2 Report
Comments and Suggestions for Authors
The study reports on the outcomes of implementing a hospitalist care system that integrates short-term admission wards with comprehensive care models to manage complex cases. Admission to the Acute Medical Unit (AMU) was associated with a lower total cost per hospital stay.
Major comments:
1. ABSTRACT: The results presented in the abstract are primarily descriptive. While most regression analyses were statistically nonsignificant, key endpoints—such as mortality, readmission, and length of stay—are important to mention. These findings emphasize the comparable care outcomes between a hospitalist-run Acute Medical Unit (AMU) and teams led by traditional attendings and internal medicine residents.
2. DISCUSSION: Please clarify the meaning of this sentence at line 248-249: “One reason may be because the teaching 248 admitter hospitalist team only covered weekdays while the admitter hospitalist team 249 covered weekends.”
Minor comments:
DISCUSSION: The 10% in-hospital mortality rate is higher than what has been reported by most hospitalist programs in the U.S. However, citing previous reports from Japan or Taiwan may help justify the effectiveness of hospitalist programs in managing complex patient populations.
Comments on the Quality of English Language
Several sentences could be made more concise. English language editing may be needed to enhance readability.
Author Response
The study reports on the outcomes of implementing a hospitalist care system that integrates short-term admission wards with comprehensive care models to manage complex cases. Admission to the Acute Medical Unit (AMU) was associated with a lower total cost per hospital stay.
Reply: We appreciate the reviewer's insightful comments and have addressed each point as follows.
Major comments:
- ABSTRACT: The results presented in the abstract are primarily descriptive. While most regression analyses were statistically nonsignificant, key endpoints—such as mortality, readmission, and length of stay—are important to mention. These findings emphasize the comparable care outcomes between a hospitalist-run Acute Medical Unit (AMU) and teams led by traditional attendings and internal medicine residents.
Reply: In this paper, we aimed to descriptively present the outcomes of the first few years of the hospitalist system. As the reviewer pointed out, mentioning the result of key endpoints between a hospitalist-run Acute Medical Unit (AMU) and teams led by traditional attendings and internal medicine residents is important. We appreciate the valuable comment and have revised the abstract accordingly. (Page 1, line 22-23)
- DISCUSSION: Please clarify the meaning of this sentence at line 248-249: “One reason may be because the teaching 248 admitter hospitalist team only covered weekdays while the admitter hospitalist team 249 covered weekends.”
Reply: We acknowledge the lack of explanation regarding this aspect and apologize regarding this aspect. This should have been addressed in the Methods section but was not adequately detailed. In our hospitalist team, the teaching hospitalist team covered patients only on workdays, whereas the admitter hospitalist team, staffed by specialists, provided coverage every day of the year, 365 days. We have now included a detailed explanation of this in the Methods section. (Page3, line 101-103)
Minor comments:
- DISCUSSION: The 10% in-hospital mortality rate is higher than what has been reported by most hospitalist programs in the U.S. However, citing previous reports from Japan or Taiwan may help justify the effectiveness of hospitalist programs in managing complex patient populations.
Reply: Our hospitalists have been found to handle more severe cases than those at any other hospitals we searched, which seems to have contributed to the higher mortality rate. As you suggested, we additionally searched previous reports from Japan and Taiwan and were able to cite references supporting the effectiveness of hospitalists in managing complex patients in the sentence on Page 9, lines 228-229. Thank you for your valuable feedback. (Page 9, line 229)
- Several sentences could be made more concise. English language editing may be needed to enhance readability.
Reply: The manuscript had been checked by a native English speaker (editage.co.kr) to address any concerns regarding the language quality of our manuscript. Although some ambiguous sentences arose during the editing process, we have carefully reviewed the paper again and made several necessary changes. We thank you for your kind comment. (Over the manuscript)
Reviewer 3 Report
Comments and Suggestions for Authors
This present study included 4,003 patients in Korea first admitted to the Hospitalist team and then discharged from the hospitalist team between March 2018 and November 2022. The study aims to show the outcomes of implementing a medical hospitalist care system.
Abstract “
Methods: This study included 4,003 patients first admitted to the Hospitalist team via ER and 12 then discharged from the hospitalist team between March 2018 and November 2022”. What does ER mean? Authors should first define before they use abbreviations.
Line 31 “Hospitalists are highly trained in managing complex medical conditions and provide around-the-clock care to patients.” Can authors explain if hospitalists are physicians and what kind of physician training they absolved?
Table 4 shows the relationship between the covariates, LOS, and the total cost per 190 hospital stay in the respective multivariate multiple regression analyses. Authors use linear regression, Did they check if length of hospital stay variable is normally distributed ? Else, linear regression is not an appropriate method.
Author Response
This present study included 4,003 patients in Korea first admitted to the Hospitalist team and then discharged from the hospitalist team between March 2018 and November 2022. The study aims to show the outcomes of implementing a medical hospitalist care system.
Reply: We appreciate the reviewer's insightful comments and have addressed each point as follows.
Abstract “ Methods: This study included 4,003 patients first admitted to the Hospitalist team via ER and 12 then discharged from the hospitalist team between March 2018 and November 2022”. What does ER mean? Authors should first define before they use abbreviations.
Reply: We apologize for the confusion regarding the use of abbreviations. We intended 'ER' to mean 'Emergency Room,' and we have corrected this in the Abstract and manuscript. Thank you. (Page 1, line 12, Page 2, line 57)
Line 31 “Hospitalists are highly trained in managing complex medical conditions and provide around-the-clock care to patients.” Can authors explain if hospitalists are physicians and what kind of physician training they absolved?
Reply: We have added clarification in the Introduction to specify that the medical hospitalist team consists of internal medicine board-certified physicians. This addition aims to prevent any confusion regarding the qualifications of the team members. Thank you for your feedback. (Page 2, line 56)
Table 4 shows the relationship between the covariates, LOS, and the total cost per 190 hospital stay in the respective multivariate multiple regression analyses. Authors use linear regression, Did they check if length of hospital stay variable is normally distributed ? Else, linear regression is not an appropriate method.
Reply: Thank you for your accurate observation. We conducted a Kolmogorov-Smirnov test to verify whether the LOS follows a normal distribution, and we confirmed that it does follows normal distribution. We appreciate your feedback.
Round 2
Reviewer 2 Report
Comments and Suggestions for Authors
The authors have improve the content of abstract and discussion section.
Minor comment:
1. The authors adress the statistical effect size of AMU regarding "Admission to the AMU was 21 associated with a lower total cost per hospital stay". This important finding was consistent with hospitalist program in most countires.
Author Response
Reviewer: 2
Minor comment:
- The authors adress the statistical effect size of AMU regarding "Admission to the AMU was 21 associated with a lower total cost per hospital stay". This important finding was consistent with hospitalist program in most countires.
Reply: Thank you for your insightful feedback. We acknowledge that there is a lack of specific studies comparing the cost differences between hospitalists managing an acute medical unit and a conventional attending and resident team. We believe this gap in the literature highlights a unique aspect of our hospitalist system, offering new insights into the cost dynamics associated with different management models within acute medical units.